# The Spanish Osteopathic Practitioners Estimates and RAtes (OPERA) study: A cross-sectional survey

Gerard Alvarez[1,2]*, Sonia Roura[1], Francesco Cerritelli[3], Jorge E. Esteves[4], Johan Verbeeck[5], Patrick L. S. van Dun[6]

1 Spain National Centre, Foundation COME Collaboration, Barcelona, Spain, 2 Iberoamerican Cochrane Centre–Biomedical Research Institute Sant Pau, IIB Sant Pau, Barcelona, Spain, 3 Clinical Human-based Research Department, Foundation COME Collaboration, Pescara, Italy, 4 Gulf National Centre, Foundation COME Collaboration, Riyadh, KSA, 5 Independent Statistical Consultant, Blaasveld, Belgium, 6 Belgium National Centre, Foundation COME Collaboration, Mechelen, Belgium

* gerardalv@gmail.com

**Data Availability Statement:** All relevant data are within the manuscript and its Supporting Information files.

## Abstract

### Background

Despite the growth of the osteopathic profession in Spain in the last few years, reliable information regarding professional profile and prevalence is still lacking. The Osteopathic Practitioners Estimates and RAtes (OPERA) project was developed as a European-based survey dedicated to profiling the osteopathic profession across Europe. The present study aims to describe the characteristics of osteopathic practitioners, their professional profile and the features of their clinical practice.

### Methods

A voluntary, validated online-based survey was distributed across Spain between January and May 2018. The survey, composed of 54 questions and 5 sections, was formally translated from English to Spanish and adapted from the original version. Because there is not a unique representative osteopathic professional body in Spain, a dedicated website was created for this study, and participation was encouraged through both specific agreements with national registers/associations and an e-based campaign.

### Results

A total of 517 osteopaths participated in the study, of which 310 were male (60%). The majority of respondents were aged between 30–39 years (53%) and 98% had an academic degree, mainly in physiotherapy. Eighty-five per cent of the respondents completed a minimum of four-year part-time course in osteopathy. Eighty-nine per cent of the participants were self-employed. Fifty-eight per cent of them own their clinic, and 40% declared to work as sole practitioner. Thirty-one per cent see an average of 21 to 30 patients per week for 46–60 minutes each. The most commonly used diagnostic techniques are movement assessment, palpation of structures/position and assessment of tenderness and trigger

**Funding:** Funding for this project was provided by the Registro de Osteópatas de España (www. osteopatas.org) grant program (ROE-2016) to SR. The funders had no role in study design, data collection and analysis, decision to publish, or preparation of the manuscript.

**Competing interests:** The authors have declared that no competing interests exist.

points. Regarding treatment modalities, articulatory/mobilisation techniques followed by visceral techniques and progressive inhibition of neuromuscular structures is often to always used. The majority of patients estimated by the respondents sought osteopathic treatment for musculoskeletal problems mainly localised on the lumbar and cervical region. The majority of respondents manifest a robust professional identity and a collective desire to be regulated as a healthcare profession.

## Conclusions

This study represents the first published document to determine the characteristics of the osteopathic practitioners in Spain using large, national data. To date, it represents the most informative document related to the osteopathic community in Spain. It brings new information on where, how, and by whom osteopathy is practised in the country. The information provided could potentially influence the development of the profession in Spain.

## Background

Osteopathy is an independent and primary contact healthcare profession recognised by the World Health Organization [1] and standardised by the European Standard EN 16686:2015 [2]. Despite health authorities' opinions, the current nation-based regulation is irregular. Indeed, Osteopathy is regulated in nine European countries (Finland, France, Iceland, Denmark, Lichtenstein, Malta, Portugal, Switzerland and the UK) and recognized in another three (Belgium, Italy and Luxembourg) [3]. Other countries such as the USA, Australia, New Zealand and Russia also have a specific regulation for the practice of osteopathy. Within this heterogeneous regulatory context, which also reflects heterogeneous training and education methods, the WHO published the *"WHO Benchmarks for training in Osteopathy"* in 2010, including a definition of the osteopathic profession, training and education criteria and ethical considerations [1] in order to define, strengthen and promote a scientific-based framework. Subsequently, the European Committee for Standardization (CEN) published in 2015 the Standards of Osteopathic Healthcare Provision which specifies requirements and recommendations for healthcare provision, facilities and equipment, education and training, and ethical framework for the practice of osteopathy [2].

In 2013, the Osteopathic International Alliance (OIA) attempted to identify the number of osteopaths working worldwide [4]. Although this was an essential preliminary stage in describing the scope and characteristics of osteopathic practice, this approach produced a significant reporting bias, particularly in those countries where osteopathy is still unregulated as is the case in Spain [5]. Within this uncertain scenario, the Spanish Ministry of Health, Social Policy and Equality published in 2011 the report "Analysis of the situation of natural therapies". This document profiled osteopathy as a complementary and alternative medicine classifying it under the category "Manipulative and body-based therapies" [6], without, however, formal recognition of the profession. This vague and imprecise legal scenario, alongside with numerous private training programmes, has resulted in the coexistence of different osteopathic profiles represented by different professional associations [7].

In recent years, some studies have been conducted in different countries to either profile the osteopathic practitioner or the osteopathic consumer [8–12]. In Spain, despite some academic dissertations addressing this issue [13–15], there is only one published study providing

this type of information [7]. However, the limitations of this study compromise the generalisability of the results.

The Osteopathic Practitioners Estimates and RAtes (OPERA) project has been developed and defined as an internationally-based survey project dedicated to profiling the osteopathic profession across Europe. It seeks to meet the need of the European community to obtain an up-to-date and reliable account regarding the geo-distribution, prevalence, incidence and profile of osteopaths in Europe. The OPERA study has been conducted in the Benelux [10] and Italy [16], updated in Belgium/Luxemburg [17] and is currently being carried out in Portugal, Austria and France [www.opera-project.org].

This study aims to profile the actual situation of osteopathy in Spain. The objective is to describe the characteristics of osteopathic practitioners, their professional profile and the features of clinical practice, regarding their practice and patient characteristics, demographic information and use of diagnostic and treatment modalities.

## Material and methods

### Objectives

The primary outcome of the present cross-sectional survey was to describe the osteopathic population in Spain, identifying their profile, clinical features and patients characteristics.

### Population

A voluntary, online-based survey was distributed across Spain between January and May 2018. Due to the variety of osteopathic training programmes available in Spain, eligible participants were required to fulfil the following inclusion criteria: "any practitioner who works in Spain and defined her/himself as osteopath, regardless of his / her education, academic degrees, whether or not different professions are combined, and where and when the training took place". Respondents had to consent their participation in the online study presentation page. By giving informed consent, they were able to access the survey. The study was approved by the Institutional Review Board of the COME Collaboration Foundation (11/2017).

### Recruitment

A dedicated website was created for this study. An e-based campaign was set up to reach the Spanish osteopathic population. Because there is not a unique representative osteopathic professional body in Spain, an online search was performed to identify all the registers, osteopathic education institutes (OEI), associations or groups of osteopaths in the country. All parties were invited by email to collaborate by spreading the information about the OPERA study to their associates. Among those who replied, an agreement form was signed and their logo was included on the OPERA website as a study partner (http://www.comecollaboration. org/es/entidades-colaboradoras/). In addition, a combined social media (Facebook, Twitter) and newsletter strategy was implemented. An e-flyer campaign was designed with nine banners published in social media during the four-month recruitment period. Three promotional videos were created to explain the objectives of the study. A newsletter campaign was established based on four emails during the four-month recruitment period and data collection. Participation in the study was completely voluntary.

### Survey tool

The OPERA study used a validated questionnaire based on the one used in the Benelux [10] survey, only adapted according to new insights and regional requirements. Adaptation was

due to the influence of a more extensive and international research team in the project and the involvement of the project-leader (PvD) in a related professional doctorate project [18]. The main adaptations introduced in the Spanish version, after having analysed the Benelux Osteo-survey, were the omission of some original questions that did not seem to contribute essential information and the omission/adaptation of others with questioned validity. Questions related to potentially sensitive information (e.g. fees) were adapted to closed-ended questions with categories. Some other open-ended questions were adapted to closed-ended after the implementation of the Benelux Osteosurvey results. Finally, a new chapter with two questions about "professional identity and views as an osteopath" was added.

The questionnaire respected the anonymity and privacy of data following the European directive 2002/58/CE of the European Parliament. The survey was translated following the forward-backward process recommended by the WHO by an English-Spanish translator with experience in demographic health research [18]. The questionnaire was composed of 54 questions and 5 sections collecting data on socio-demographics, osteopathic training, working profile, organisation and management of clinical practise and patient profile. A pre-pilot study was conducted on 20 Spanish-speaking osteopaths to validate the questionnaire.

An OPERA survey online platform, already developed and used, and an implemented data warehouse utilised for research purposes was used for this study [16]. The data entered was encrypted and sent via internet using an ad-hoc software named COME Survey developed explicitly to run highly secure surveys and studies containing potentially sensitive data [16]. This system transfers data to a certified data centre; all information is processed and hosted the following data protection regulations. Answers were anonymised, and IP addresses were not disclosed to the research team. The system automatically manages the link between email address, Study ID, and survey status, which means that research staff was not able to identify the responses provided. Only OPERA research personnel had access to the complete, anonymised dataset.

## Information guidelines

Participants were asked to complete the forms by completing the information regarding demographics, working status and professional activities, education, consultation fees, patient complaints, treatment and management.

## Statistical analysis

The sample size was arbitrarily predicted and calculated assuming all practitioners that were granted a diploma in osteopathy from the Spanish OEI from their inception to May 2018. This information was asked to the OEI (regardless if they accepted or not formal participation in the study), producing an estimated sample of 5,427 osteopaths. Considering a standard deviation of 10%, it was predicted that the number of osteopaths in Spain ranged from 4,800 to 5,900.

Taking into account that the survey response rate varied between 10 and 60% of those receiving the questionnaire [19], the number of practitioners who would participate at the survey was estimated between 480 to 3,540. This calculation also considered all the osteopaths who graduated in a foreign country. Furthermore, all the information gathered was analysed and reported as grouped data. Completed questionnaires were individually examined and no attempt was made to identify respondents. Results are presented following a descriptive analysis using frequencies and percentages for qualitative variables. R statistical programme (v. 3.1.3) was used.

## Results

### Osteopaths

A total of 517 osteopaths participated in the study, of which 310 were male (60%). Fifty-three per cent of the respondents were aged between 30–39 years, followed by 32% that were between 40–49 years old (**Table 1**). There is a gender shift in the 20–29 years age category (**S1 Table**). Almost half of the participants (46%) completed their osteopathic education and training in the 5 years before completion of this study. Distribution of respondents was all around the country although higher participation was detected in Catalonia (37%), followed by Madrid Community (14%) and the Basque Country (10%) (**S1 Fig**). Less than half of respondents (47%) are members of or registered in one of 18 associations and registers in Spain (**S2 Table**).

### Work status, setting and activities

Eighty-nine per cent of the respondents were self-employed, 58% of them owned their clinic and 40% declared to work as sole practitioners (**Table 2**). Among those who stated to work with other professionals, physiotherapists (29%) or other osteopaths (28%) were the most common colleagues, followed by dieticians (8%), podiatrists (8%) and psychologists (7%) (**Fig 1**). More than half of the respondents (61%) reported having other professional activities apart from their clinical practice as osteopaths (**S3 Table**). Respondents declared referring patients to other professionals, as shown in **Fig 2**. **Fig 3** represents the frequency of the referrals received by Spanish osteopaths. In 47% of the cases, respondents have interest in treating specific patient groups, such as children (17%), infants (15%), pregnant women (15%), athletes (15%), patients with specific pathologies (12%) and patients with gynaecological conditions (12%). Concerning their consultation policy, the majority of respondents informed patients about data protection policy (78%), confidentiality policy (80%), chaperone policy for minors (75%) and for treating intimate zones (74%) and cancellation policy (39%).

### Osteopathic training and lifelong learning

Almost all respondents had a prior academic degree (98%), mainly in physiotherapy (75%), (**S4 Table**). The majority of respondents (81%) completed their osteopathic education and training in Spain, 5% abroad and 14% combined. The most common type of training among respondents lasted a minimum of four years part-time (85%). Of all respondents, 81% attended continuous professional development courses (CPD) during the previous year. The majority of them, over 25 hours (84%).

**Table 1. Age and gender of respondents (n = 517).**

| Gender | N | % |
|---|---|---|
| Male | 310 | 59.9 |
| Female | 207 | 40.1 |
| **Age** | N | % |
| 20–29 | 51 | 9.8 |
| 30–39 | 276 | 53.3 |
| 40–49 | 163 | 31.5 |
| 50–59 | 19 | 3.6 |
| 60–65 | 6 | 1.1 |
| >65 | 2 | 0.3 |

**Table 2. Working status of respondents (n = 517).**

| Type of employment | N | % |
|---|---|---|
| Self-employed | 460 | 88.9 |
| Employed | 57 | 11.0 |
| **Type of clinical condition** | N | % |
| Clinic owner | 268 | 58.2 |
| Business partner of a clinic | 167 | 36.3 |
| Associate | 25 | 5.4 |
| **Type of working collaboration** | N | % |
| Alone | 211 | 40.8 |
| In group | 197 | 38.1 |
| Both | 109 | 21.0 |

## Professional identity

Respectively 85% and 78% of respondents agreed or strongly agreed with the statements *"I strongly define myself as a healthcare practitioner"* and *"I strongly define myself as an osteopath"*. Also, 54% of respondents agreed or strongly agreed with the statement "*Medical professionals see osteopathy as a distinct healthcare discipline"*. Being regulated (80%), the possibility that patients could receive a better reimbursement (93%) and the willingness to better collaborate with other healthcare professionals (94%), were also important concerns among respondents. Eighty-five per cent of participants agreed or strongly agreed that *"regulation would have a positive effect on how osteopaths practice"* and 92% that "osteopathy *should be regulated as a first-line medical practice"*. Seventy-one per cent agreed or strongly agreed with the statement *"osteopathy should be regulated as a paramedical profession"*. Surprisingly, only 57% per cent of respondents agreed or strongly agreed that *"overall, the quality of patient care provided by osteopaths in Spain is good"*.

## Consultation structure

The majority of respondents worked five days per week in clinical practice (60%). Main practice characteristics are described in **Table 3**.

## Patients

While 51% of respondents declare that their patient database is equally balanced between men and women, 32% report that it's mostly women who go to see them. Patients ranged between 18–40 years of age are treated very often by 67% of the respondents and 36% reported to never treat patients younger than one month (**Fig 4**). According to respondents, the majority of patients consulted them for musculoskeletal problems, mainly of the lumbar and cervical regions. **Table 4** shows the type of complaints most frequently treated by respondents. Respondents confirm that they are consulted almost evenly 'very often' for chronic (55%) and acute (53%) problems over the last year. Furthermore, osteopaths refer that prevention is also a common cause of consultation (**S5 Table**).

## Treatment and diagnosis

Over 64% of respondents performed an osteopathic assessment at every consultation, while 35% confirmed to perform it regularly to often but not always. Exclusion diagnostics (i.e. diagnosis of a medical condition reached by process of elimination when presence cannot be

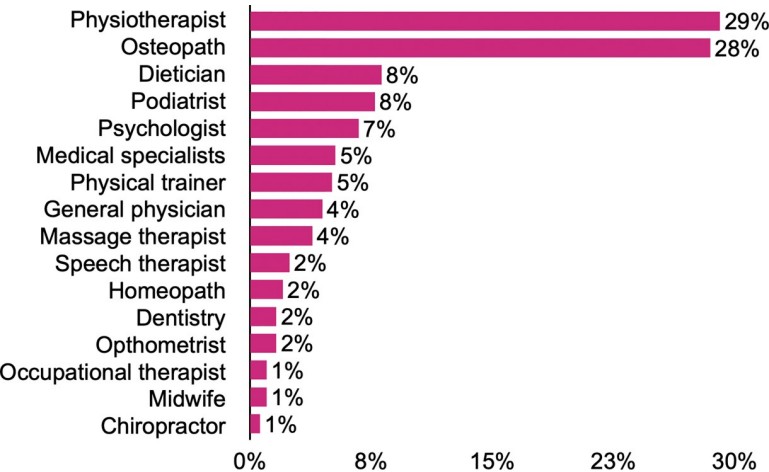

**Fig 1. Distribution (%) of osteopath's working colleagues.**

established with complete confidence from history, examination or testing), was always performed by 59% of the respondents and 99% declared informing patients about possible risks and secondary reactions to treatment and their possible benefits (**Table 5**). The most frequently used diagnostic techniques can be found, in decreasing order, in **Table 6** and treatment approaches in **Table 7**. Of all techniques applied to internal and sensitive areas, intraoral techniques were the most used (36% "often" and 20% "always"). Internal genital and rectal techniques are frequently used ("often" or "always") by 16% and 9% of respondents respectively. Informed consent for this type of technique was requested by 78% of respondents. Within the recommendations given as part of the treatment plan, physical activity and advice on exercises were the most common among respondents (94%). The main reasons for referring patients to other healthcare professionals were *"not my field of expertise"* (66%) and if there were *"indicators of undiagnosed pathology or structural deficit"* (65%). *"Increase in level of*

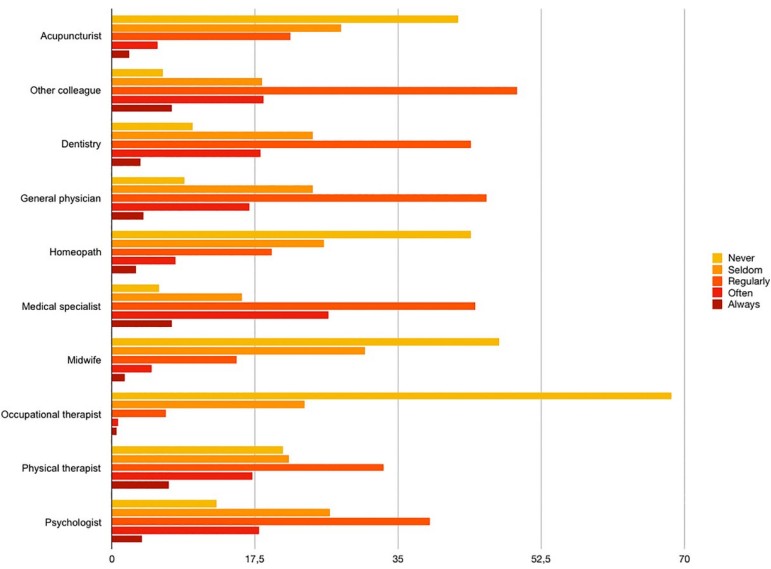

**Fig 2. Modalities of referring patients from osteopaths to another professional (%).**

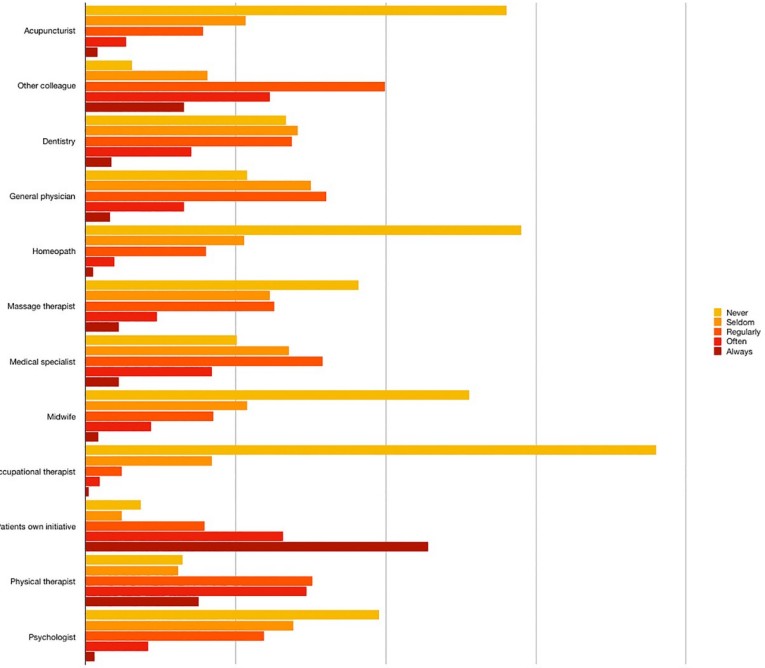

**Fig 3. Modalities of referring patients from other professionals to osteopaths (%).**

*primary symptoms"* (58%) was the third main reason for referring patients to other healthcare professionals (**S6 Table**).

## Discussion

In general, the typical osteopath in Spain is male, aged between 30–39 years, with a previous academic degree, mainly in physiotherapy, and part-time osteopathic education. These results are in line with the only previous study published in Spain [7], with the Benelux Osteosurvey [10] and the Italian OPERA version [16]. However, the latest 2011 KPMG Report profiling osteopaths [19] showed that UK osteopaths are older (41–50). As in Italy and the Benelux, the results in Spain showed a process of feminisation of the osteopathic profession with a gender shift in the 20–29 years age category (**S1 Table**). This is a process already evidenced in other countries like Australia, where a recently conducted workforce survey showed that 58% of respondents were female [12]. Moreover, in countries like Switzerland, almost two-thirds of osteopaths are female [20][21]. In fact, women's participation is expanding in traditionally male-dominated professions [22]. Although there is still no evidence on the reasons behind this shift and how feminization impacts the quality of care in manual therapy disciplines, it has been shown that in medicine, the quality of care provided by women may result in improved population health [23].

### Osteopathic training

Regarding osteopathic training, results showed similar trends in Spain and the Benelux both with educational programmes commonly lasting a minimum of 4 years in a part-time format and after a prior degree (mainly in physiotherapy). However, in Italy, osteopaths commonly follow a 6-year part-time educational programme and the majority of them hold a previous

**Table 3. Main practice characteristics (n = 517).**

| Consultation time for new patient | | | Consultation time for returning patient | | |
|---|---|---|---|---|---|
| Time | N | % | Time | N | % |
| <30 min | 0 | 0.0 | <30 min | 6 | 1.1 |
| 30–45 min | 49 | 9.4 | 30–45 min | 124 | 23.9 |
| 46–60 min | 319 | 61.7 | 46–60 min | 363 | 70.2 |
| >60 min | 149 | 28.8 | >60 min | 24 | 4.6 |
| **Number of patients on consultation a week** | | | **Number of new patients on consultation a week** | | |
| Patients | N | % | Patients | N | % |
| 0–10 | 64 | 12.4 | 0–5 | 347 | 67.1 |
| 11–20 | 117 | 22.6 | 6–10 | 135 | 26.1 |
| 21–30 | 162 | 31.3 | 11–15 | 25 | 4.8 |
| 31–40 | 112 | 21.7 | 16–20 | 7 | 1.3 |
| 41–65 | 58 | 11.2 | > 20 | 3 | 0.5 |
| > 65 | 4 | 0.7 | | | |

| Average waiting period for first consultation | | | |
|---|---|---|---|
| Patients | | N | % |
| Same day | | 19 | 3.6 |
| Next business day | | 51 | 9.8 |
| Within 2–7 business days | | 302 | 58.4 |
| Within 8–14 business days | | 84 | 16.2 |
| Between 2–4 weeks | | 34 | 6.5 |
| > 1 Month | | 27 | 5.2 |

| Fee first consultation | | | Fee following consultation | | |
|---|---|---|---|---|---|
| € | N | % | € | N | % |
| <25 | 10 | 1.9 | <25 | 11 | 2.1 |
| 26–40 | 155 | 30.0 | 26–40 | 200 | 38.7 |
| 41–60 | 280 | 54.2 | 41–60 | 270 | 52.2 |
| 61–80 | 62 | 12.0 | 61–80 | 34 | 6.6 |
| 81–100 | 8 | 1.5 | 81–100 | 1 | 0.2 |
| >100 | 2 | 0.4 | >100 | 1 | 0.2 |

academic degree in sports science or physiotherapy. In Switzerland, only about one-third of osteopaths had a previous degree [20].

The publication of the document "Benchmarks for training in Osteopathy" by the World Health Organization [1] in 2010 and especially the approval of the European Standard EN 16686:2015 in 2015 by the European Committee for Standardization [2] constitute the first solid steps to reach a common training framework on the continent. However, the lack of specific European regulation on training before the publication of these documents has led to the existence of professionals trained in a wide variety of osteopathic educational programmes.

## Practice characteristics and professional fees

Of all respondents, 89% (n = 462) were self-employed, with 58% owning their practice. Again, these results agree with the ones found in Italy and the Benelux [10,16].

Forty per cent of respondents state to work as a sole practitioner and 38% work in a group practice. Spanish, Benelux and Swiss osteopaths follow a similar trend (34% in Benelux and 39.9% in Switzerland) [10,21] whereas most of Australian osteopaths work in group practices (83.7%) [12]. Spanish and Benelux osteopaths tend to work mostly with other osteopaths in

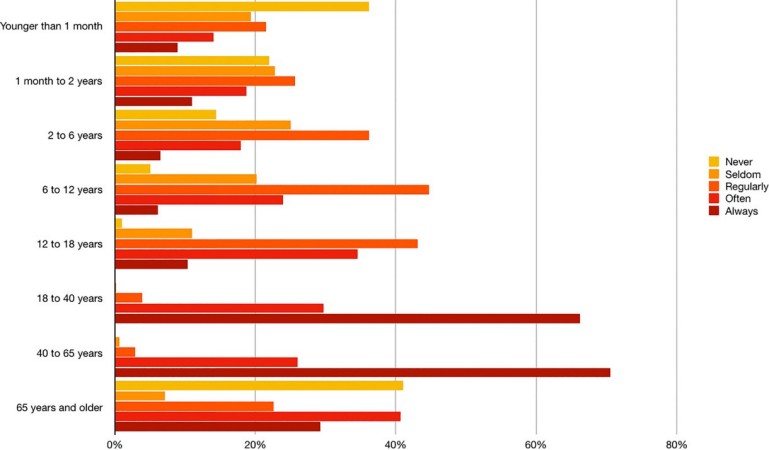

**Fig 4. Patient's age distribution (%).**

the same practice. However, almost 60% of Italian and 57% of UK osteopaths declared to work alone in their practice [16,19]. Regular referrals are standard practice among Spanish respondents. However, patients in Spain mainly consult an osteopath through self-referral (**Fig 4**) in the pursuit of an alternative to previous unsuccessful treatments [7]. In this context it should be highlighted that interprofessional care is one of the ingredients for building effective modern health systems [24] and, although efforts have been made to promote interprofessional care within osteopathy, this interest should extend to research [25]. According to Alvarez et al., up to 69% of patients had previously visited a physician for the same complaint with an average number of 2.8 consultations before consulting an osteopathic practitioner. This data highlights the lack of a bidirectional professional relationship between osteopaths and other healthcare practitioners in Spain. Several reasons can explain this situation. First, as previously mentioned, osteopathy lacks formal regulation in Spain and is therefore mainly practised within the private healthcare sector, compromising both the access of lower socio-economic groups to this form of healthcare provision and the normal patient-flow between professionals. For example, in countries where osteopathy is formally recognised and regulated by law (i.e. Australia), osteopaths and other health professions declared receiving referrals from each other regularly [12]. Secondly, the coexistence of numerous qualifications and professional

**Table 4. Specific type of complaints (Ten most common in descending order adding "often" and "always" responses).**

|  | never | seldom | regularly | often | always |
|---|---|---|---|---|---|
| back pain | 0.3 | 0.5 | 4.6 | 32.5 | 61.9 |
| neck complaints | 0.3 | 1.9 | 9.0 | 35.5 | 53.0 |
| sciatica | 0.9 | 3.0 | 13.1 | 38.6 | 44.1 |
| headache and migraine | 0.7 | 3.4 | 18.3 | 47.7 | 29.5 |
| cervicobrachialgia | 0.5 | 6.3 | 25.1 | 42.5 | 25.3 |
| craniomandibular complaints | 3.0 | 14.3 | 36.1 | 29.5 | 16.8 |
| complaints during / after pregnancy / childbirth | 10.8 | 17.7 | 30.1 | 26.1 | 15.0 |
| digestive disorders | 3.8 | 17.2 | 40.0 | 23.9 | 14.8 |
| baby colic | 29.4 | 18.5 | 19.5 | 16.8 | 15.6 |
| shoulder problems | 8.5 | 26.8 | 34.8 | 20.8 | 8.9 |

Numbers in table are %

**Table 5. Frequency of performance of osteopathic activities over the last year in clinical practice.**

| | never | seldom | regularly | often | always |
|---|---|---|---|---|---|
| inform about previous treatment | 0.1 | 0.5 | 4.2 | 17.4 | 77.5 |
| inform possible risks and secondary reactions to treatment | 0.3 | 1.3 | 3.6 | 20.7 | 73.8 |
| inform possible benefits of treatment | 0.1 | 0.7 | 3.4 | 22.0 | 73.5 |
| explain treatment plan | 0.1 | 0.3 | 4.2 | 22.4 | 72.7 |
| consider patients history | 0.1 | 0.7 | 4.6 | 22.2 | 72.1 |
| examination at every consultation | 0.5 | 0.7 | 9.0 | 25.5 | 64.0 |
| patient reaction to treatment | 0.1 | 0.9 | 8.1 | 29.2 | 61.5 |
| perform exclusion diagnostics to determine treatment | 0.9 | 2.3 | 11.9 | 25.3 | 59.3 |
| inform about other treatments or consequence of no treatment | 2.1 | 5.2 | 19.7 | 32.6 | 40.2 |
| inform about patients expectations of treatment | 2.5 | 7.1 | 22.0 | 35.0 | 33.2 |

Numbers in table are %

associations representing different groups of osteopaths undermines the profession's coherence and credibility. Finally, osteopathy is a relatively young profession in Spain (dating from the 1980s) [26] and, despite the considerable development in recent years [7], it is still unknown to many healthcare professionals.

Spanish respondents organise their consultations in a similar way to UK osteopaths and colleagues from the Benelux. As a common pattern in all those countries, osteopaths plan 46–60 minutes for a first consultation and between 30–45 minutes for follow-up visits [10,19,27–30]. German osteopaths tend to opt for a longer consultation time [10]. Consultation fees in Spain also seem to be in line with osteopaths from the Benelux (**Table 3**) [10].

Although some private health insurances include osteopathy in their services, in most cases, patients are predominantly private and pay out-of-pocket the cost of the treatment [7], drawing attention to the problem of access to osteopathic care for lower socio-economic groups.

**Table 6. The most common diagnostic techniques used (in descending order adding "often" and "always" responses).**

| | never | seldom | regularly | often | always | Don't know |
|---|---|---|---|---|---|---|
| palpation of movement | 0.3 | 1.3 | 3.6 | 17.9 | 75.8 | 0.7 |
| palpation of position/structures | 0.3 | 1.5 | 5.6 | 17.7 | 73.8 | 0.7 |
| tender points and trigger points | 0.9 | 2.3 | 6.3 | 19.1 | 70.7 | 0.3 |
| assessment of visceral mobility | 1.7 | 6.7 | 15.2 | 23.6 | 52.2 | 0.3 |
| assessment of the cranium (neuro- and viscerocranium) | 3.4 | 5.2 | 15.8 | 23.9 | 50.2 | 1.1 |
| visual inspection | 6.7 | 8.1 | 11.6 | 12.5 | 58.9 | 1.9 |
| muscle function testing | 1.7 | 9.0 | 18.1 | 30.1 | 40.0 | 0.7 |
| imaging | 5.4 | 7.1 | 20.8 | 35.2 | 28.0 | 3.2 |
| neurologic testing | 5.0 | 11.4 | 21.2 | 31.5 | 28.4 | 2.3 |
| fascial testing | 7.5 | 11.6 | 19.5 | 24.7 | 34.0 | 2.5 |
| orthopedic testing | 13.9 | 12.3 | 22.4 | 20.1 | 24.3 | 6.7 |
| percussion and auscultation | 13.1 | 15.4 | 25.9 | 19.7 | 21.2 | 4.4 |
| neurolymphatic reflex tests | 20.7 | 16.6 | 18.7 | 17.7 | 15.4 | 10.6 |
| otoscopy | 21.8 | 20.5 | 22.8 | 16.0 | 10.6 | 8.1 |
| blood analysis | 30.1 | 20.7 | 19.7 | 9.8 | 4.4 | 15.0 |
| urine testing | 37.7 | 19.7 | 14.1 | 4.0 | 2.5 | 21.8 |

Numbers in table are %

**Table 7. The most common therapeutic techniques used (in descending order adding "often" and "always" responses).**

|  | never | seldom | regularly | often | always | Don't know |
|---|---|---|---|---|---|---|
| Articulatory/mobilisation techniques (GOT/TBA) | 2.7 | 5.0 | 6.7 | 20.1 | 63.8 | 1.5 |
| visceral manipulations | 2.3 | 6.0 | 12.3 | 27.8 | 51.2 | 0.1 |
| Progressive Inhibition of Neuromuscular Structures (PINS) | 2.3 | 4.4 | 14.5 | 26.6 | 50.8 | 1.1 |
| neurocranial and viscerocranial techniques | 4.0 | 5.0 | 12.7 | 24.1 | 53.3 | 0.5 |
| functional techniques | 1.9 | 5.6 | 15.2 | 26.3 | 50.6 | 0.1 |
| soft and connective tissue techniques | 3.6 | 6.1 | 13.3 | 26.8 | 49.1 | 0.7 |
| HVLA techniques | 6.0 | 6.7 | 10.4 | 21.4 | 53.0 | 2.3 |
| fascial techniques | 4.6 | 9.8 | 19.7 | 25.1 | 38.8 | 1.7 |
| MET | 7.1 | 10.6 | 19.5 | 27.4 | 31.1 | 4.0 |
| fluid techniques | 6.0 | 13.5 | 27.6 | 30.1 | 19.9 | 2.7 |
| automatic shifting and fluid body approach | 15.6 | 12.9 | 16.4 | 14.3 | 13.5 | 27.0 |

Numbers in table are %

GOT/TBA "General Osteopathic Treatment / Total Body Adjustment"

HVLA "High Velocity Low Amplitude"

MET "Muscular Energy Techniques"

This situation could explain the fact that more than 65% of the respondents stated that they apply a fee reduction for economically challenged patients. Despite the evidence about the effectiveness of osteopathy to treat highly prevalent pain conditions such as back pain, sciatica or neck pain [31] the integration of osteopathy in the delivery of healthcare services generally remains unclear [25]. We argue that a better integration could potentially benefit the Spanish population. As an example, in Catalunya it is considered that 20% of the population choose complementary and alternative therapies to treat their pain issues [32].

### Patients, diagnostic and therapeutic modalities

According to respondents, the typical patient in Spain is middle-aged and seeking osteopathic care for predominantly acute or chronic musculoskeletal problems, mainly localised in the lumbar and cervical region. This is in agreement with the data reported in previous studies in Spain [7,33], and data obtained in the Benelux [10], UK [9], Quebec [8], Australia [11,12] and Switzerland [21].

In some parts of the survey, participants were asked to rate their response according to a "frequency of use" scale ranging from "never" to "always" or "very often" with an additional "don't know" option. Although this system allows for an accurate interpretation of the frequency in which respondents use some procedures, it makes it difficult to rank them from most to least used. For this reason, despite the system used in the Benelux Osteosurvey [10], the results were presented with decreasing order of those with higher frequencies of "often" and "always" responses instead of a mean value. The results obtained on diagnostic and treatment procedures (**Tables 6 and 7**) are in agreement with the data found by Alvarez et al. [7] except for the high rate response regarding the visceral techniques. While in this previous Spanish survey just 28% of respondents declared to use visceral techniques in their treatment (7th most frequently used osteopathic technique), up to 80% of respondents in the current study declare to use them often or always (2nd most frequently used technique modality). Differences can be explained by the way this specific question was asked in each survey. In the study of Alvarez et al. osteopaths were asked to rank from most-to-less used treatment techniques and separately the first and follow-up consultations. In the current study, as mentioned

above, different "frequency of use" options were available to respondents for each treatment technique. This way of asking could also explain the differences found in the use of High-Velocity Low Amplitude (HVLA) techniques between both surveys. Considering the results of the two surveys, it can be argued that either visceral or HVLA techniques are frequently simultaneously applied and combined with other techniques, which are how osteopathic treatments are commonly delivered in clinical practice. Notwithstanding this, the higher use of visceral and cranial techniques was a result also noticed and discussed in the Benelux Osteosurvey [10] compared to other similar studies. As the authors argued, differences in the use of technique modalities between countries (mainly UK and Australian osteopaths) can be related to the fact of having or not a prior degree in physiotherapy and the need to distinguish osteopathic practice from other manual health professions [34]. For example, according to the last KPMG-study conducted in the UK [19], 32% of respondents declared to never use visceral manipulation and 37% declared to spend from 0–10% of the time per week using that approach. Similar trends are shown for cranial techniques (respectively 26% and 22%). In our study, 77% of respondents declared to perform neurocranial and viscerocranial techniques often (24%) or always (53%). Another example is Australia, where just 10% of respondents declared to use visceral manipulation and 23.5% cranial manipulation as treatment modalities [12][35]. Interestingly, a recently published secondary analysis shows associations between the use of visceral techniques and a range of practice characteristics of Australian osteopaths, including engaging with research to inform practice [35]

From the beginning, the osteopathic profession in Spain was related to physiotherapy. The earliest educational programmes developed in Spain were designed as postgraduate training for physiotherapists. Nowadays, different types of training and professional profiles coexist. Osteopaths with a prior physiotherapy degree constitute the largest group of professionals. Relatively speaking, some similarities can be found in the development of osteopathy in Australasia and the existence of dual-qualified osteopath/chiropractors [36]. Unfortunately, to date, there are no exact and reliable data on the number and profiles of practising osteopaths in Spain. However, taking our sample as an indicator of the global situation in the country, it can be said that around 75% of the osteopaths have a prior degree in physiotherapy. Forty years after the beginning of the osteopathic profession in Spain, the relationship between both professions is controversial. The scenario is polarized between those who consider osteopathy as an exclusive competence of physical therapists and those who consider osteopathy as an independent healthcare profession and pursue its regulation. The only legal text mentioning osteopathy is the Ministerial Order (2135/2008) from the Law of Arrangement of the Health Professions (LOPS 44/2003) that establishes the training curriculum of the physiotherapy degree. In this text, osteopathy is mentioned as a technique that undergraduates shall know [37]. In this survey, 80% of respondents agree or strongly agree with the statement that "*osteopathy should be regulated as an independent healthcare profession*". Although 74% recognises that other healthcare professionals in Spain perceive osteopathy as a manual therapy subgroup. Altogether, our results show a shared feeling of professional identity and a collective desire to become a distinct healthcare profession.

## Limitations

Some limitations should be considered when interpreting our data. The first limitation of the study is related to sample size which can be biased by either the number of OEI that accepted to provide data or by the reliability of the data. In addition, the response rate couldn't be calculated due to the fact that the invitation to participate was not made personally. However, the number of responses obtained are sufficient to verify if there is a difference between categories

[38]. In fact, our numbers are similar or superior to those found in other surveys conducted among osteopaths in the UK [39], Quebec [8] or even similar to recent surveys conducted among physiotherapists in Germany [40], Saudi Arabia [41] or Australia [42]. Only the completion of a regulatory process and the creation of the mandatory official register will allow to exactly know the number of Spanish osteopaths. Secondly, practitioners were responsible for data entry and therefore, results could be affected by respondent bias. Thirdly, more than half of the results were obtained from osteopaths located in two specific regions in Spain (Catalonia and Madrid). Although these results can easily be explained by demographic and academic reasons, this can again influence the interpretation of the data and not be representative of the entire osteopathic profession in Spain.

## Conclusions

The OPERA-ES study represents the first published document to determine the characteristics of osteopathic practitioners in Spain using an extensive, national sample. This study, alongside with a previous publication focused on the patient's profile [7], provides for the first time a comprehensive dataset describing the osteopathic scenario in Spain. Our findings could have implications for the development of the profession. First, it represents the most informative document related to the osteopathic community in Spain, bringing new information on where and how osteopathy is practised, and how osteopaths profile themselves. Secondly, comparisons between countries were discussed in order to highlight differences across European and with the rest of the world. Finally, the information provided could contribute to the body of evidence used by stakeholders and policymakers in a potential future regulation of the profession in Spain.

## Supporting information

**S1 Fig. Distribution (%) of participants across the different Spanish regions.**
(DOCX)

**S1 Table. Relation age and gender among respondents.**
(DOCX)

**S2 Table. Osteopathic associations and registers (alphabetic order).**
(DOCX)

**S3 Table. Other professional activities (n = 517).**
(DOCX)

**S4 Table. Academic degrees.**
(DOCX)

**S5 Table. Patients reason for consultation.**
(DOCX)

**S6 Table. Reasons for referring patients.**
(DOCX)

**S7 Table. Consultation policy.**
(DOCX)

**S8 Table. Content of osteopathic training.**
(DOCX)

**S9 Table. Osteopathic identity.**
(DOCX)

## Acknowledgments

The authors wish to acknowledge in first place all the surveyed participants. Secondly, the following Research Collaborative Partners: Registro de los Osteópatas de España (ROE), Federación de Osteópatas de España (FOE), Instituto Español de Osteopatía Clásica (IEOC), Escuela de Formaciones Osteopáticas (EFO Madrid), Formación Belga Española de Osteopatía (FBEO), Escuela de Osteopatía de A Coruña (EOC) European Osteopathy and Research Academy (EORA), Escola d'Osteopatia de Barcelona (EOB), Escuela de Osteopatía y Técnicas para la Salud (EOTS), Instituto de Osteopatía de Valencia (INOVA), Asociación de Profesionales de Osteopatía (APREO), Sociedad Europea de Medicina Osteopática (SEMO), Escuela de Técnicas Naturales (ESTENA), Asociación Profesional Española de Terapias Naturales (APROE-TENA), Academia de Osteopatía Craneal (AOC). Finally, authors also wish to thank Dr. Gines Almazán.

## Author Contributions

**Conceptualization:** Gerard Alvarez, Sonia Roura, Francesco Cerritelli, Jorge E. Esteves, Johan Verbeeck, Patrick L. S. van Dun.

**Data curation:** Francesco Cerritelli, Patrick L. S. van Dun.

**Formal analysis:** Johan Verbeeck.

**Funding acquisition:** Gerard Alvarez, Sonia Roura.

**Investigation:** Gerard Alvarez, Sonia Roura.

**Methodology:** Gerard Alvarez, Sonia Roura, Francesco Cerritelli, Jorge E. Esteves, Patrick L. S. van Dun.

**Project administration:** Gerard Alvarez, Sonia Roura, Francesco Cerritelli, Jorge E. Esteves, Patrick L. S. van Dun.

**Resources:** Gerard Alvarez, Sonia Roura, Francesco Cerritelli, Jorge E. Esteves, Patrick L. S. van Dun.

**Software:** Francesco Cerritelli.

**Supervision:** Gerard Alvarez, Sonia Roura, Francesco Cerritelli, Jorge E. Esteves, Johan Verbeeck, Patrick L. S. van Dun.

**Validation:** Gerard Alvarez, Sonia Roura, Francesco Cerritelli, Jorge E. Esteves, Johan Verbeeck, Patrick L. S. van Dun.

**Visualization:** Gerard Alvarez, Sonia Roura.

**Writing – original draft:** Gerard Alvarez, Sonia Roura.

**Writing – review & editing:** Gerard Alvarez, Sonia Roura, Francesco Cerritelli, Jorge E. Esteves, Johan Verbeeck, Patrick L. S. van Dun.

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
