## [Decision Letter · Decision Letter 0]

4 May 2020

PONE-D-20-08905

The Spanish Osteopathic Practitioners Estimates and RAtes (OPERA) study: A cross sectional survey

PLOS ONE

Dear Alvarez,

Thank you for submitting your manuscript to PLOS ONE. After careful consideration, we feel that it has merit but does not fully meet PLOS ONE’s publication criteria as it currently stands. Therefore, we invite you to submit a revised version of the manuscript that addresses the points raised during the review process.

We would appreciate receiving your revised manuscript by Jun 18 2020 11:59PM. To enhance the reproducibility of your results, we recommend that if applicable you deposit your laboratory protocols in protocols.io, where a protocol can be assigned its own identifier (DOI) such that it can be cited independently in the future. For instructions see: http://journals.plos.org/plosone/s/submission-guidelines#loc-laboratory-protocols

We look forward to receiving your revised manuscript.

Kind regards,

Jenny Wilkinson, PhD

Academic Editor

PLOS ONE

Reviewers' comments:

Reviewer's Responses to Questions

**Comments to the Author**

1. Is the manuscript technically sound, and do the data support the conclusions?

Reviewer #1: Yes

Reviewer #2: Partly

2. Has the statistical analysis been performed appropriately and rigorously? 

Reviewer #1: N/A

Reviewer #2: Yes

3. Have the authors made all data underlying the findings in their manuscript fully available?

Reviewer #1: Yes

Reviewer #2: Yes

4. Is the manuscript presented in an intelligible fashion and written in standard English?

Reviewer #1: Yes

Reviewer #2: Yes

5. Review Comments to the Author

Reviewer #1: This study contributes to the data needed on osteopaths within different countries. Meanwhile participation rate was low, recruitment process was difficult with no official registry. The main comment that I would transmit is that authors wanted to show all the data collected in the manuscript. Moreover, there are a lot of tables and figures (9 tables and 5 figures). I would suggest authors to think about reducing the data shown, in order to help readers to focus on the most important data. Some of the data should be transferred in the supplemental material. For example, the figure about the regions where respondents practiced does not bring many information to the readers, as it is not representative of the prevalence of osteopaths in each region of Spain.

Abstract :

In the methods, I would suggest to have more elements described in the methods of the abstract: questionnaire already used in other studies, translated into Spanish,… (elements of what is written in the survey tool part of the manuscript).

It seems that there is no registry of all osteopaths within Spain. For this reason, the authors reached the potential participants through different ways. This should be clarified shortly in abstract. This should be also clarified in the manuscript. For example, it would be interesting to know how many osteopaths were reached by emails (even if a part of participants were not reached by emails).

Manuscript:

Introduction:

The questionnaire should be better described: was the questionnaire changed since the first version (I think it was van Dun et al. State of affairs of osteopathy in the Benelux: Benelux Osteosurvey 2013)? If yes, what was changed?

The authors mentioned that “A secondary objective is to compare the results with other similar studies to establish potential differences across countries. » I am not sure that it can be an objective, as the authors did not analyze the data from other countries. They compared their results with other countries, which is not an objective, but one part of the discussion.

Methods:

Survey tool: it is mentioned that the questionnaire was adapted to “new insights and regional requirements”. I would recommend that the authors describe more precisely what was changed in the questionnaire.

Statistical analysis: If my understanding is right, diploma in osteopathy are delivered by private schools. I do not understand why inclusion criteria was not based on the practitioners who had a diploma, but on practitioners who considered themselves as osteopaths. I suggest to explain better the reason for this decision.

“Statistical analyses were based on a univariate and multivariate approach. R statistical programme (v. 3.1.3) was used.” I do not find any univariate or multivariate data. It is not clear why this sentence was mentioned. On the other hand, it would be probably interesting to know if there are some differences linked with sex and age of osteopaths (having another professional activity, number of patients each week, …).

Results:

Tables: I suggest to write only one number after the comma.

Table 2: I do not think that this table is useful. I would suggest to remove it.

Figure 1. I do not think that this figure is useful, as it represents the respondents and not the number of osteopaths for each region. I would suggest to remove it.

Supporting information S1: promotion videos is a very good idea. Meanwhile, I am not sure that it will useful to give the access to readers in the manuscript.

Professional identity: I find this part very important and interesting. I would suggest to have a figure with some of theses results. On the other hand, I would recommend to authors decide if data in figure 3, 4 and 5 should really be in a figure.

Table 5. I would suggest to reduce the number of groups for “number of patients on consultation each week”. It is hard to read now. Same for fee.

Discussion:

“Although respondents come from almost all Spanish regions, the higher concentration of practitioners is located in Catalunya and Madrid (figure 1), which are the autonomous communities with the largest population in Spain.” As this result could be linked with the recruitment process, I would suggest to remove this sentence.

It would be interesting if you also compare your results with data from Switzerland:

https://journals.plos.org/plosone/article?id=10.1371/journal.pone.0224098

I would suggest, where possible, to reduce the length of the discussion.

Conclusion:

The conclusion should be better written, only linked with what the data says. For example, I am not sure that the study estimated the prevalence of osteopaths. It also did not use representative data. Moreover, it does not give reliable information on where osteopathy is practiced.

“Finally, the information provided can be used by stakeholders and policymakers in a potential future regulation of the profession in Spain.” I would remove this sentence. Other studies are needed to get sufficient reliable data.

Reviewer #2: Thank you for the opportunity to review this manuscript. I provide some comments that I hope the authorship team will find useful.

Overall, this manuscript could benefit from another close copyedit to strengthen the written English/grammar. Some examples are provided below, but a full review is needed.

Background:

P6: I think the term "heterogenous" would suit better than "inhomogenous"

P6: Also, do you mean training and education methods? rather than processes?

P7: Please avoid paragraphs of only two sentences. The two paragraphs on this page (starting "In 2013..." and "Within this uncertain...") can be edited to form one more cohesive paragraph.

P7: edit - "is still unregulated as is the case in Spain"

P8: edit - "numerous training programs has resulted in the coexistence of..."

P8: in what way does the unpublished studies provide unreliable information?

Materials and methods:

Please replace all numbers less than 10 with the words (e.g. ten) where they occur in sentences.

P10: in the abstract you refer to the prevalence of Osteopaths in Spain as part of your aim but it is not listed in the main methods section. I don't think you can confer prevalence from your methodology anyway.

P12: how was the survey adapted? what was the process you undertook to do this?

Results:

Unless it is a journal requirement, I don't think you need to provide any more than one decimal place for most of your data. It adds nothing of value to the paper and distracts from the findings in the tables and text.

Although this is a style preference, you could try reworking some of your sentences so that you are not starting so many with a numeric value. This will reduce the overall manuscript length as you will then be able to convert quite a few words to numbers.

P19: I am not sure I agree that "specific pathologies" and "gynecological conditions" count as a population group? Could this be reworded?

Table 3: The last three percentages have a comma "," in stead of a decimal "."

P21: Based on your data, does this mean some self-identifying as osteopaths may have no osteopathic qualifications at all? How would they have been trained? I understand the challenges of an unregulated environment, but this warrants some close attention in the discussion (I think).

Table 4: Even though the frequency is higher, I recommend moving "other activities" to the end of the list in this table.

P22: It is unsurprising that such a high number "strongly define as an osteopath" given those who don't would not have likely responded to the survey. More interesting is the 15% who completed the survey but do NOT strongly identify as an osteopath.

P22: The percentage of osteopaths agreeing that osteopathy should be regulated as a paramedical profession is repeated. Once in the sentence and once in parentheses at the end.

Table 5: The value of so many categories of patient numbers is not apparent. Could you collapse down some categories? Same for the number of new consultations per week. There are such lower numbers in 16-20 and >20 categories that they could easily be combined. And again for the fee categories.

P28: The sentence beginning "Increase in level of primary symptoms" does not need to be separate. It can be included in the previous paragraph.

Discussion:

My most substantial comment for this manuscript relates to the discussion. It feels unecessarily long. I think that the authors could identify 3 to 5 (maximum) of the most important discussion points and provide a nuanced engagement with those topics. As it stands, the manuscript overviews numerous findings and compares them superficially to existing research without providing the reader with any in-depth consideration of the context, meaning and application of the results.

One such example is the possible feminisation of the workforce. More discussion about why this may be occuring would be great. As another example, the point about the alignment between osteo and physio could be paralleled with the osteo/chiropractic links in Australia. The finding about low frequency of cranial/visceral is also interesting. Is that a result of a shift in evidence priorities? Training and curriculum? Ultimately the discussion would be strengthened by providing a deeper engagement with the implications of the findings rather than simply comparing them with other studies.

Conclusions:

Once the discussion has been revised, the conclusion will also need attention as it currently makes claims that are not substantiated within the study such as "offers compelling data in order to better assess better the inclusion of osteopathic services within the Spanish healthcare system". For a claim like this to be upheld your discussion will need to engage with this point directly. Also, I am not sure you can use the phrase "nationally representative" as you don't have a basis for comparison. You can say it draws on a national sample, though.

6. PLOS authors have the option to publish the peer review history of their article (what does this mean?). If published, this will include your full peer review and any attached files.

Reviewer #1: Yes: Pierre-Yves Rodondi

Reviewer #2: Yes: Amie Steel

---

## [Author Response · Author response to Decision Letter 0]

27 May 2020

A "Response to Reviewers" file with detailed response of each comment has been uploaded.

---

## [Editor Report · Decision Letter 1]

2 Jun 2020

The Spanish Osteopathic Practitioners Estimates and RAtes (OPERA) study: A cross-sectional survey

PONE-D-20-08905R1

Dear Dr. Alvarez,

We are pleased to inform you that your manuscript has been judged scientifically suitable for publication and will be formally accepted for publication once it complies with all outstanding technical requirements.

With kind regards,

Jenny Wilkinson, PhD

Academic Editor

PLOS ONE
---

## [Editor Report · Acceptance letter]

4 Jun 2020

PONE-D-20-08905R1 

The Spanish Osteopathic Practitioners Estimates and RAtes (OPERA) study: A cross-sectional survey 

Dear Dr. Alvarez:

I'm pleased to inform you that your manuscript has been deemed suitable for publication in PLOS ONE. Congratulations! Your manuscript is now with our production department. 

Kind regards, 

on behalf of

Prof. Jenny Wilkinson 

Academic Editor

PLOS ONE